# Regeneration of the Damaged Parts with the Use of Metal Additive Manufacturing—Case Study

**DOI:** 10.3390/ma16103772

**Published:** 2023-05-16

**Authors:** Piotr Sawczuk, Janusz Kluczyński, Bartłomiej Sarzyński, Ireneusz Szachogłuchowicz, Katarzyna Jasik, Jakub Łuszczek, Krzysztof Grzelak, Paweł Płatek, Janusz Torzewski, Marcin Małek

**Affiliations:** 1Institute of Robots & Machine Design, Faculty of Mechanical Engineering, Military University of Technology, Gen. S. Kaliskiego St. 2, 00-908 Warsaw, Poland; piotr.sawczuk@student.wat.edu.pl (P.S.); bartlomiej.sarzynski@wat.edu.pl (B.S.); ireneusz.szachogluchowicz@wat.edu.pl (I.S.); katarzyna.jasik@student.wat.edu.pl (K.J.); jakub.luszczek@wat.edu.pl (J.Ł.); krzysztof.grzelak@wat.edu.pl (K.G.); janusz.torzewski@wat.edu.pl (J.T.); 2Institute of Armaments Technology, Faculty of Mechatronics, Armaments and Aerospace, Military University of Technology, Gen. S. Kaliskiego St. 2, 00-908 Warsaw, Poland; pawel.platek@wat.edu.pl; 3Institute of Civil Engineering, Faculty of Civil Engineering and Geodesy, Military University of Technology, Gen. S. Kaliskiego St. 2, 00-908 Warsaw, Poland; marcin.malek@wat.edu.pl

**Keywords:** additive manufacturing, laser-based powder bed fusion of metals, selective laser melting, tensile strength, M300 maraging steel, regeneration

## Abstract

The paper shows the results related to regeneration possibilities analysis of a damaged slider removed from a hydraulic splitter that was repaired using additive manufacturing (AM), laser-based powder bed fusion of metals (PBF-LB/M) technology. The results demonstrate the high quality of the connection zone between the original part and the regenerated zone. The hardness measurement conducted at the interface between the two materials indicated a significant increase equal to 35% by using the M300 maraging steel, as a material for regeneration. Additionally, the use of digital image correlation (DIC) technology enabled the identification of the area where the largest deformation occurred during the tensile test, which was out of the connection zone between the two materials.

## 1. Introduction

Nowadays, researchers are currently investigating effective methods to repair damaged machinery components, particularly those that are subject to high loads in hydraulic systems used in modern machinery constructions. Small components within these systems are particularly susceptible to stress and damage. To address this, new methods for the production and regeneration of damaged components are being explored, with a focus on AM technologies. One of the ways of new production and regeneration of damaged components is AM technologies, that allow for the production of many components with complex geometries, in many cases impossible to manufacture using conventional methods such as casting or subtractive machining [1,2,3]. The significant advantage of these technologies is the ability to produce lightweight structures with low mass while maintaining the required strength properties [4,5]. AM processes have the advantage of being able to use various metals and their alloys [6]. Additionally, the AM process allows for cost reduction by properly screening unused material and using it in subsequent processes [7]. What is more, the AM also allows for cost reduction by effectively utilizing unused materials in subsequent processes. Furthermore, during the initial phase of AM, appropriate manufacturing parameters such as laser power and material layer thickness can be selected to significantly influence the mechanical properties of the final product [8,9,10,11]. These benefits have led to an increase in scientific research and publications on the application of additive manufacturing techniques, particularly in the production and regeneration of machine parts and equipment. One example demonstrating the potential of additive manufacturing is the attempts at the regeneration of crankshafts [12,13]. In the work of H. Koehler et al. [13], laser-directed energy deposition (L-DED) technology was used to regenerate the worn surface of the crankshaft by cladding additional layers of material. The authors obtained an HV0.1 microhardness growth in the connection zone between the original and parent material from 300 to 450 HV0.1, and further growth in the heat-affected zone to 500 HV0.1. Additive techniques are also used to regenerate all types of gears [14,15,16]. An interesting approach related to the regeneration of damaged parts has been shown in the research conducted by Zhu et al. [14,15]. In both works, the authors used a five-axis industrial robot equipped with the 500 W fiber laser cladding tool. In the first approach related to gear tooth regeneration [14], a significant change in the connection zone between the original and cladded material was registered. In the region of the molten pool, the authors obtained columnar dendrites, that show and grow epitaxially perpendicular to the substrate surface. In the second article related to the laser cladding of Inconel 718 [15], the authors tried to improve the regeneration based on the laser cladding by using ultrasound during this process. Such kind of approach allowed for obtaining microstructural changes, based on columnar crystal concentration changes which led to microhardness improvements. Additive manufacturing techniques have also been used to regenerate boats and ship propellers [17,18]. Taşdemir et al. carried out an analysis using wine arc additive manufacturing (WAAM) [17], in the regeneration of ship parts, based on the available state of the art. The authors concluded that WAAM technologies have significant potential for parts remanufacturing, but there is an important need for simplification and improvements of slicing algorithms in the dedicated software. On the other hand, Montero et al. [19] used the selective laser melting (SLM) technology to re-design the valve cover, but without using the original material. A few interesting examples demonstrating the potential of implementation of the AM techniques are described in [20,21,22,23]. One of the analyzed works conducted by S.K. Rittinghaus et al. [20] involved the use of a titanium aluminum alloy to produce a component that satisfactorily passed a tensile test where 1200 MPa yield strength was achieved. Based on the literature survey related to the regeneration of machine parts, it has been observed that a significant number of the papers are concerned with the regeneration of rotor blades used in aerospace industries, such as jet engine turbines and turbochargers. Additionally, most of the case studies have been made with the use of DED technologies. 

Repairing damaged parts often involves the necessity of replacing the original parts with components made from available materials that fill the strength requirements such as titanium alloys (e.g., Ti6Al4V), Inconel alloys, or maraging steels. However, identifying the exact native material can be difficult and time consuming. Therefore, it is essential to evaluate the mechanical and microstructural quality of the deposited layer of materials which may differ significantly in terms of chemical composition and mechanical properties. Although there are numerous examples of the application of AM technologies in the repairing process of machinery and equipment, nevertheless there is a lack of research on their application in regenerating hydraulic system components. Furthermore, noteworthy is the significant popularity of utilizing maraging M300 steel in the production of various components using additive manufacturing techniques. However, there is a lack of detailed information and guidelines regarding the process of deposition of maraging M300 steel on the other materials. In this work, the results of the regeneration process of a slider from a hydraulic splitter using the SLM method and maraging M300 steel are described. Based on the performed mechanical strength tests, the evaluation carried out on the quality of the joint made in comparison to the original part is included in this paper.

## 2. Materials and Methods

A component that was subjected to the aforementioned regeneration process was a slider removed from a hydraulic splitter. Based on the geometrical measurement results carried out with the use of basic workshop tools, a 3D CAD model of the slider was designed in Solidworks 2022–2023 software (SolidWorks Corp., Waltham, MA, USA). To identify the chemical composition of the native material used to produce the slider, a portable Olympus Delta DPO-2000CC spectrometer was used. It uses a built-in database with exact patterns that allow for the determination of the content of individual alloying components of the analyzed material. The measurements were carried out at a temperature of 20 °C and a humidity level of 33% at two points on the shaft presented in Figure 1. The chemical composition data registered during this measurement are also shown in Table 1. 

In this study, the mechanical strength of a repaired component using additive manufacturing techniques was verified through the intentional damage of the component by cyclic loading to simulate fatigue, followed by regeneration and assessment of its impact on mechanical properties. In the initial stage of the study, a hydraulic machine was used to cyclically load the slider. To simulate the conditions of slider failure, the element was stretched using an Instron 8802 testing machine until it broke. To regenerate a broken part using the SLM125HL printer, the working chamber dimensions had to be taken into consideration. Therefore, the broken part was torn in a way that allowed for proper installation in the chamber. The surface of the part needed to be aligned perpendicular to the slider axis and subjected to additional abrasive flow machining to ensure better adhesion of the printed material to the native material. Next, the remaining part was measured and a computer model was created for the missing fragment of the slider. The model was slightly modified compared to the original dimensions, mainly by increasing individual dimensions to allow for additional machining to obtain a surface with lower roughness. A comparison between the model and actual dimensions, along with any excess material, is presented in Figure 2.

Proper performing of the regeneration process of the damaged element required an additional mounting device that allows an auxiliary orientation of the regenerated slider in alignment with the building plate. For this purpose, a dedicated holder was developed in CAD—SolidWorks 2022–2023 system and manufactured using a desktop Fused Filament Fabrication 3D printer (3DGence, Przyszowice, Poland) and PET-G filament. An additional holder allowed for the proper leveling of the powder layers and a tight fit of the slider to prevent uncontrolled movements during the regeneration process. The manufacturing process of the regenerated slider was proceeded by a preparation program controlling this process. The parameters such as laser power, hatch distance, exposure velocity, and layer thickness were identified to provide the most favorable properties of the produced element. The identified parameter values are presented in Table 2. After attaching the platform with the regenerated element to the machine’s movable table and lowering it, the space was filled with maraging M300 steel powder. The result of the work is visible in Figure 3. 

To begin, approximately 60 layers of maraging M 300 steel were deposited based on the 3D printing process onto the non-regenerated portion of the slider that remained after it broke. This kind of approach has been used to assess the initial quality of the connection between the deposited layers of maraging M300 steel and the original material. The regenerated part has not been subjected to any additional heat treatment. The regenerated part was subsequently used for a metallographic cross-section analysis. This analysis was used to conduct microhardness measurements at the interface between the two materials. The slider that has been regenerated through the SLM process is displayed in Figure 4A. In order to achieve a proper alignment for further tensile tests in uni-axial conditions, the produced part has been additionally machined, as shown in Figure 4B. 

The regenerated component was subjected to static tension tests. The universal strength machine Instron 8802 (Instron, Norwood, MA, USA) and the digital image correlation (DIC) (Dantec, Ulm, Germany) method were employed to precisely analyze the deformations that took place in the material during the test. Additionally, the fractured element was analyzed at a macroscopic level to identify the areas where the stress propagated and to examine the material’s structure. To accomplish this, a Keyence VHX7000 digital microscope, as illustrated in Figure 5, was utilized. Microstructure analysis was conducted on both the original and regenerated materials, with a particular focus on the junction between the two materials, by utilizing higher magnification values. The analysis was performed using scanning electron microscopy (SEM) with an energy dispersive spectroscopy (EDS) system, which provided a detailed assessment of the chemical composition of the materials at their interface. This enabled a more precise determination of the individual alloying elements present in each of the steels. Furthermore, microhardness measurements of the joint were conducted in accordance with the PN-EN ISO 6507-1 standard [24], utilizing the Struers DuraScan hardness tester shown in Figure 6.

## 3. Results and Discussion

### 3.1. Preliminary Chemical Composition Analysis of the Material

The chemical composition analysis of the material allowed for an estimation of the material used to manufacture the slider. The measurement results indicated the use of high-quality non-alloyed steel P235GH, which is primarily utilized for producing parts that operate at high temperatures. This type of steel offers excellent plasticity, bending strength, and welding properties, making it suitable for manufacturing machine components such as rotor shafts in turbochargers or devices that operate at high pressures. Table 3 presents the measurement results of the individual element content, along with their respective normative values according to EN10253-2. It should be noted that when measuring a larger diameter part of the shaft, a high chromium content (Cr = 33%) was observed. This value is most likely attributed to surface treatment to increase its wear resistance.

### 3.2. Static Tensile Test with Additional Use of the DIC Method

A tensile test (shown in Figure 7) was conducted on both the original and regenerated reference slider samples using a universal testing machine. The original sample fractured in the area of its smallest diameter equal to d = 9.4 mm. The dedicated machine’s software controlling the test allows for registration deformation history plot which is presented in Figure 8. The maximum stress value reached during the tests was approximately 1020 MPa, and the sample fractured at a strain equal to 3.5%. Using the DIC method, areas of the sample with the largest deformations were identified. Figure 8 displays the sample’s deformation process at two critical points, namely during the peak stress period (Figure 8A) and at the point of slider fracture (Figure 8B). In the left image, warmer colors such as yellow and orange highlight the areas experiencing the most substantial deformations, whereas, in the image on the right, these same regions coincide with the site of fracture.

The subsequent test was carried out to examine the mechanical response of the regenerated slider subjected to tension loading conditions. Adopted initial boundary conditions were similar to the previous studies. As a result, the nominal stress–strain curve was registered. It is presented in Figure 9A. The maximum value of nominal stress was approximately 450 Mpa. Application of the DIC system during the tension test allowed for the identification of the regions of slider geometry which demonstrated the highest value of deformation and initialization of the material fracture. Due to the unfavorable camera placement during the test, only a fragment of the area of the greatest deformation is visible in the image presented in Figure 9B. The most important conclusion from this image is that the fracture did not occur at the junction of the two materials. The greatest deformations were recorded in the area of the printed part made from M300 steel, in a place located about 1 mm above the beginning of the additionally designed fillet.

### 3.3. Macro and Microstructural Analysis

The next stage of the conducted research was related to the macroscopic and microscopic evaluation of the fracture surfaces of the samples after the strength tests. A Keyence VHX7000 digital microscope with a lens allowing observation at different magnification levels was used for this purpose. Based on the obtained images of the fracture surfaces, it was found that in the case of the original slider, the fracture surface has a plastic character in the central part and a brittle character in the outer part (Figure 10). This differentiation is due to the applied heat treatment aimed at increasing the hardness of the outer surface directly subjected to the load and reducing its susceptibility to wear. The plastic character of the core slider fracture is intended to ensure its resistance to fatigue loads. The regenerated version of the slider demonstrates a uniform fracture across its entire surface, suggesting a brittle fracture (Figure 11). 

A detailed evaluation of the surface of a slider variant subjected to a regeneration process revealed a material structure characterized by the presence of inclusions in the form of loose grains of powder of varying sizes that were not melted during the process.

By analyzing the appearance of the regenerated slider, the location of its fracture, and its microstructure, it can be concluded that several layers shifted during the start of printing a larger diameter shaft. The most likely cause of this phenomenon could be the lack of support structures in areas with geometric overhangs, which were not possible to generate because of the usage of polymeric substrate plates.

High magnification (×100) was utilized to examine the connection between the original material of the slider and that deposited with the use of a 3D printing technique. Figure 12 presents an image of the area where the maraging M300 steel is deposited on the original surface of the slider which indicates a ferritic–pearlitic structure. The presence of a martensitic structure close to the pin’s surface indicates that surface hardening was used to increase the resistance to wear. The slider’s core, on the other hand, remained unhardened, which results in higher resistance to dynamic loads. Upon analyzing the images that depict the quality of the deposited material, it becomes apparent that the material contains certain structural imperfections such as voids that have an irregular shape and porosity that may have been caused by gas becoming trapped during the manufacturing process. Additionally, the outer walls of the material exhibit partially melted powder particles. 

The microstructure of the area where the native material is joined with the regenerated material is depicted in the following images. Figure 13 shows visible traces of the boundaries of local melt pools resulting from the use of selective laser melting (SLM) additive manufacturing technology. Additionally, there are irregularly shaped voids that represent structural defects in the material. However, after a thorough microscopic examination of the joint between the native and SLM material, no significant structural defects were found that could affect the mechanical properties of the joint. To ensure a better adhesive bond with the applied layer of material, the front surface of the regenerated element was subjected to sandblasting.

It should be highlighted that there are no defects, such as bubbles or non-metallic inclusions, present throughout the entire length of the connection in this area. Upon analyzing the entire surface of the connection of both materials, no micro cracks that could negatively affect strength were detected. The images presented in Figure 14 indicate the presence of a mixing zone between the materials, which is a desirable phenomenon observed along the entire length of the connection.

After analyzing Figure 14, it becomes evident that the transitional zone formed due to the elevated temperature during the process has a directional structure of fine-needle martensite.. The zone initially has a residual amount of ferrite and is approximately 50–60 µm wide. This zone has a gradient character and transitions smoothly from the native material to the regenerated material without any visible defects, which is visible in Figure 15.

### 3.4. Chemical Composition Analysis Using SEM

Detailed analysis of the tested object, including its elemental composition, was conducted using scanning electron microscopy (SEM) with an energy dispersive spectroscopy (EDS) system. To analyze the chemical composition between two materials, the linear EDS measurement (Figure 16) was conducted. The graphs presented below (Figure 17, Figure 18, Figure 19 and Figure 20) illustrate changes in the content of the analyzed elements along the length, expressed in units of “cps” (counts per second), as the number of counts per second. The measurement was obtained over a total length of approximately 550 µm, as shown in the diagram (Figure 16). 

Upon analysis of Figure 17, it can be observed that the Fe content is higher on the side of the native material. The contents of key elements, including Ni (Figure 18), Cr (Figure 19), and Mn (Figure 20), which is part of the composition of the maraging steel (M300) used, were also analyzed. According to the specified chemical composition for standard non-alloyed steel, these elements exist within a limited range. The graphs generated from the research results confirm this relationship.

The transitional zone between the native material and the maraging M300 steel applied using an additive manufacturing technique is particularly significant. Based on the conducted analysis of the chemical composition, it can be observed that the amount of individual alloying additions changes continuously depending on the considered area. The heat generated during the melting and mixing of both materials in this zone had a significant influence as well.

### 3.5. Hardness Measurement

To compare the functional properties of the slider after regeneration, hardness tests were carried out. The measurements were made using the Vickers scale with a load of 1 kg (HV1) based on the PN-EN ISO 6507-1 standard. Figure 21 shows the sample view after hardness measurements in each zone. Three measurement series were carried out in each zone: A—measurement at the joint of the native material with the M300 material, B—measurement in the transition zone, and C—measurement in the heat-treated zone of the native material’s surface layer.

The hardness of both the base material and the AM-regenerated part is represented by the measurement conducted at the joint between these elements. Three series of measurements (A1, A2, A3) were performed on a measurement section of 3 mm, starting from the M300 material side. The results of each measurement were registered. Based on the obtained results, it was possible to visualize these data on the chart in Figure 22. Analyzing the hardness profiles as a function of distance, it can be stated that the regenerated part is characterized by higher hardness. The results of each series exhibit good repeatability. The point in the joint area included in the A3 series clearly shows the highest value. This result may be related to the strengthening mechanism typical for maraging steel, i.e., the precipitation of intermetallic phases in the martensitic matrix [25,26]. The average hardness of the regenerated layer presented in the chart is about 390 HV, while the average hardness of the base material is about 260 HV. However, it should be remembered that each series of measurements was performed in an area not covered by the heat-treatment zone.

Additionally, an interesting issue was to investigate the hardness in the transition zone where both materials are joined. For this purpose, three measurement series were additionally carried out. The number of indentations was limited due to the size of the zone. It is worth noting the hardness results obtained at the center of the boundary between both materials (Figure 23). Their average value is 450 HV, which is about 60 HV higher than in the case of the regenerated material.

To illustrate the effect of heat treatment on hardness, three measurement series were carried out in the zone near the outer surface towards the core of the reference slider. The results of one of the measurement series are presented in the HV1 hardness graph as a function of distance. Analyzing the hardness course (Figure 24), it can be concluded that in the heat-treated zone, the hardness increases to a value of approximately 620 HV.

## 4. Conclusions

Based on the obtained results, the following conclusions were drawn:During the tensile test, the SLM-regenerated hydraulic splitter slider achieved a stress level of 450 MPa. In the same test, the original slider achieved a value of 1000 MPa. However, the failure of the regenerated sample did not occur at the connection between the two materials, but rather in the 3D-printed material.The connection between the parent and regenerated material achieved a hardness of 630 HV. The maraging steel M300, which constituted the regenerated material, achieved a hardness of 390 HV, while the native material achieved 260 HV.The good quality of the connection was confirmed based on microstructural analysis. The two materials were properly mixed. No defects were observed along the entire length of the connection.The M300 steel meets the parameters dictated by the properties of the reference part. This confirms the quality of the connection, as well as the results of the strength and hardness tests.

## Figures and Tables

**Figure 1 materials-16-03772-f001:**
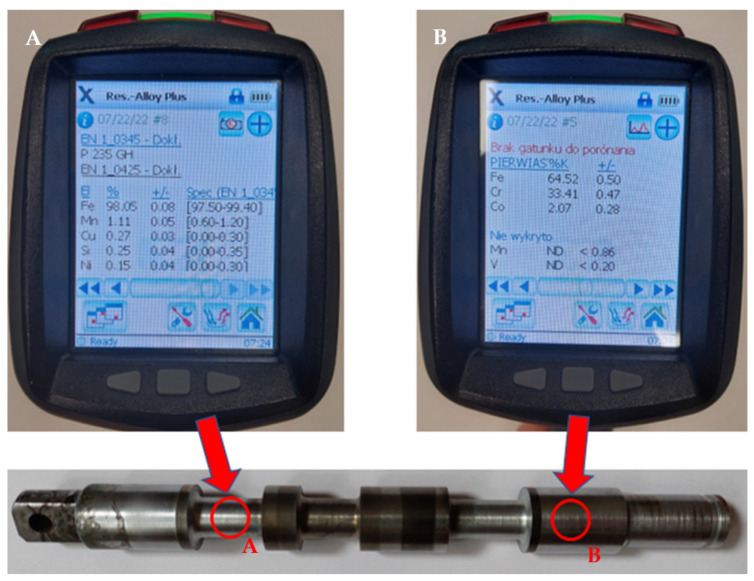
Chemical composition analysis on the hydraulic splitter slider. (**A**) surface without chrome—plating, (**B**) chrome-plated surface.

**Figure 2 materials-16-03772-f002:**
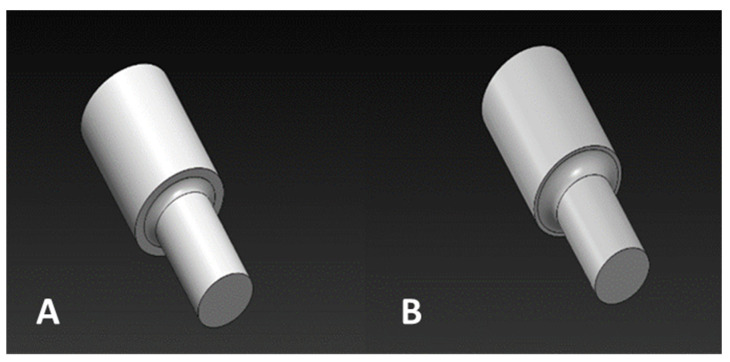
Three-dimensional CAD model selected for regeneration: (**A**) before modification; (**B**) after modification with an increased fillet radius.

**Figure 3 materials-16-03772-f003:**
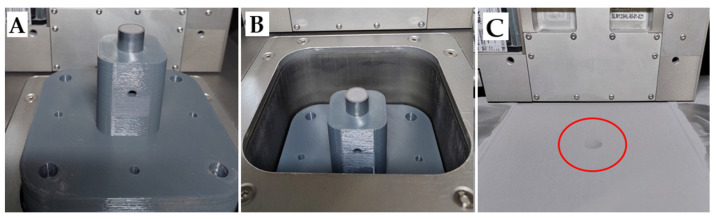
(**A**) View of the base attached to the working platform; (**B**) view of the lowered platform below the zero level; (**C**) view of the base surface alignment with the zero level–visible top surface of the original part (marked by the red circle).

**Figure 4 materials-16-03772-f004:**
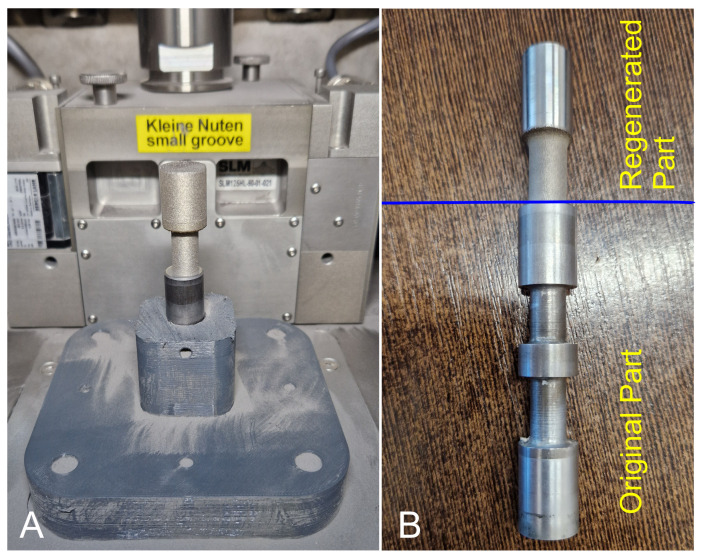
(**A**) Regenerated part after SLM process; (**B**) whole part after regenerated machining.

**Figure 5 materials-16-03772-f005:**
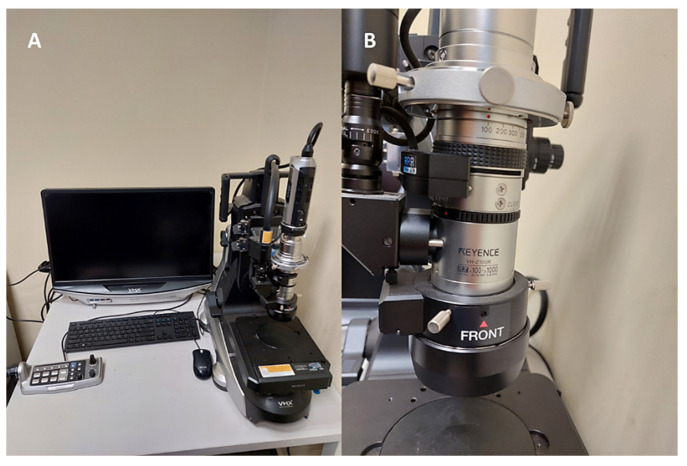
(**A**) Keyence VHX7000 digital microscope; (**B**) lens used for taking photos of the fractures.

**Figure 6 materials-16-03772-f006:**
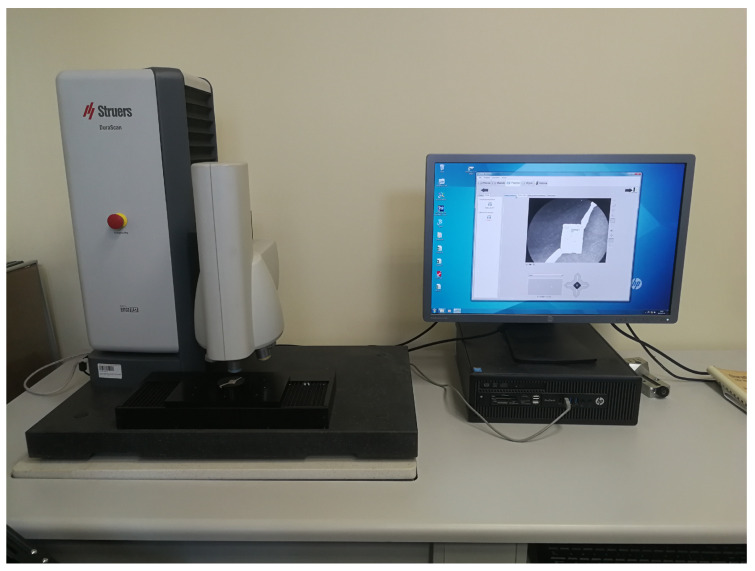
The Struers DuraScan hardness tester used during the research.

**Figure 7 materials-16-03772-f007:**
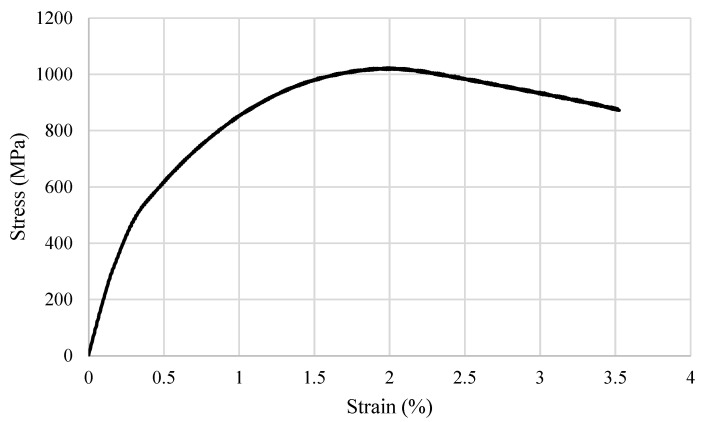
Mechanical response of original slider subjected to tensile test with the results of DIC measurements carried out during tests.

**Figure 8 materials-16-03772-f008:**
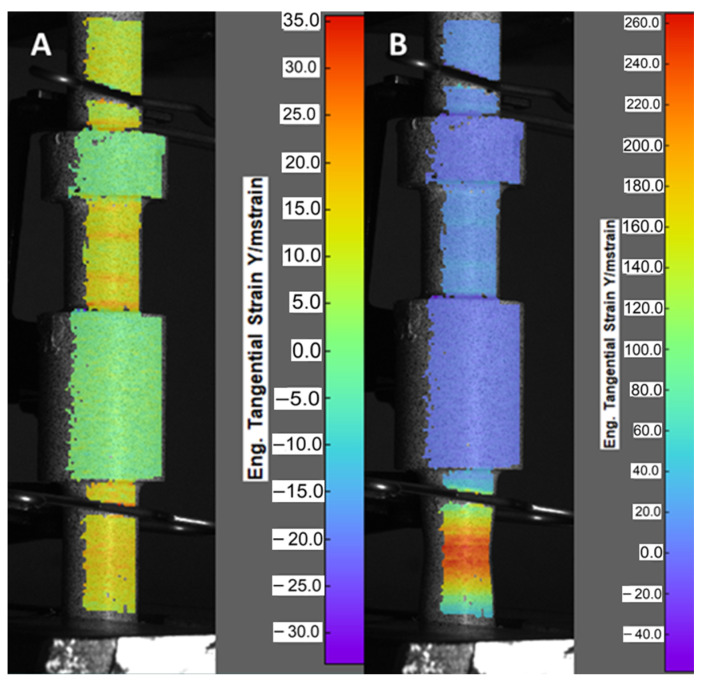
(**A**) DIC image of the reference slider at the moment of loading causing stress equal to the tensile strength (UTS); (**B**) DIC image of the reference slider just before the fracture.

**Figure 9 materials-16-03772-f009:**
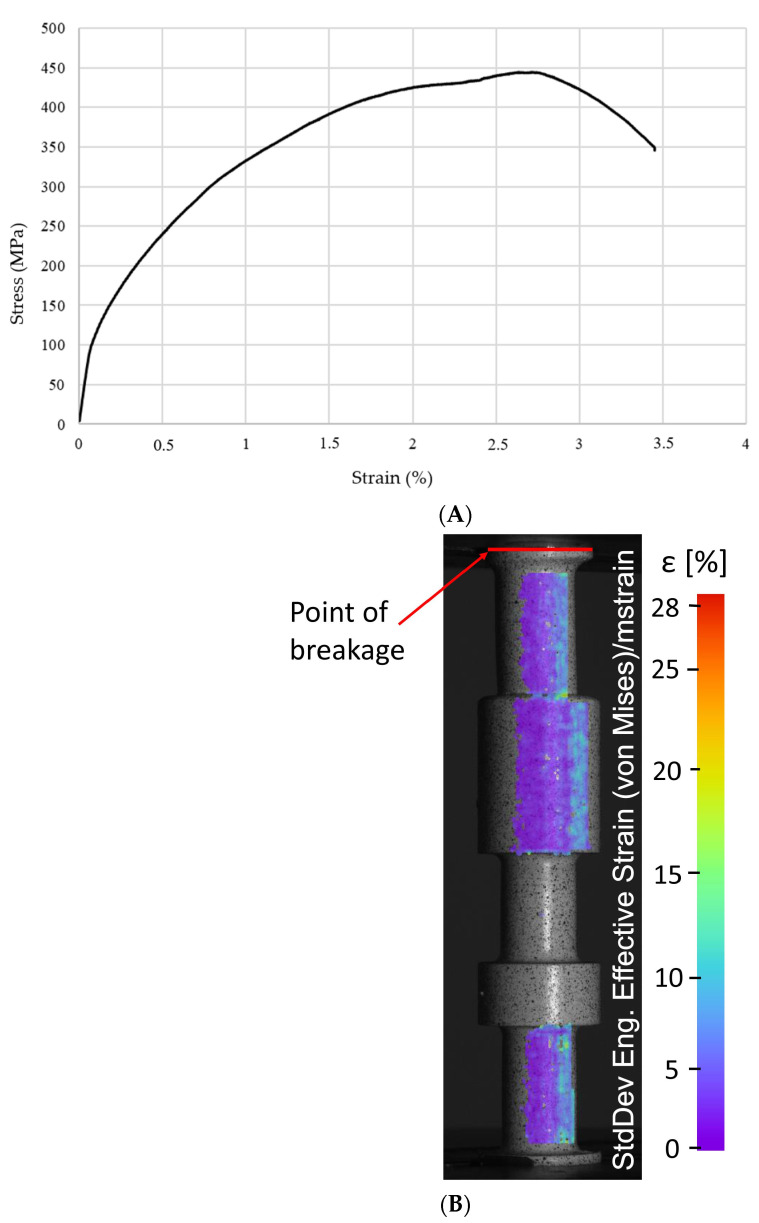
Mechanical response of regenerated slider subjected to tensile test: (**A**) nominal stress-strain plot, (**B**) results of DIC measurements carried out during tests.

**Figure 10 materials-16-03772-f010:**
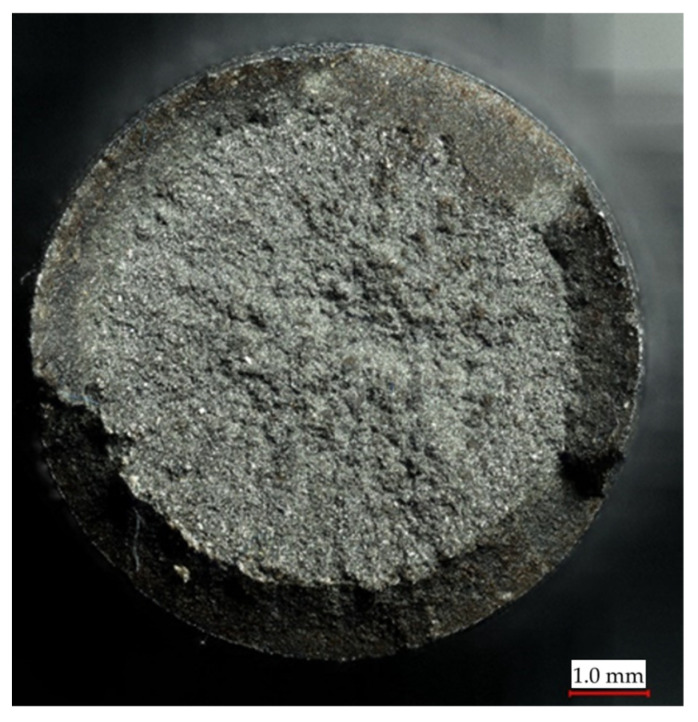
An image of the fracture surface of the reference slide after a quasi-static tensile test.

**Figure 11 materials-16-03772-f011:**
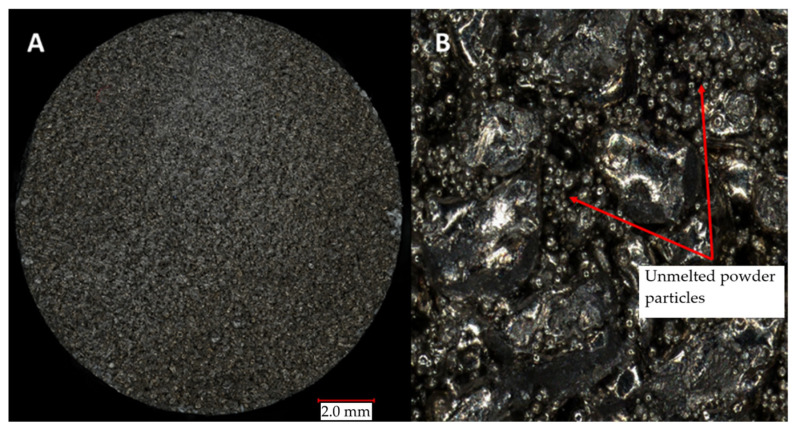
(**A**) View of the fracture of the regenerated slider; (**B**) enlarged view of the fracture surface.

**Figure 12 materials-16-03772-f012:**
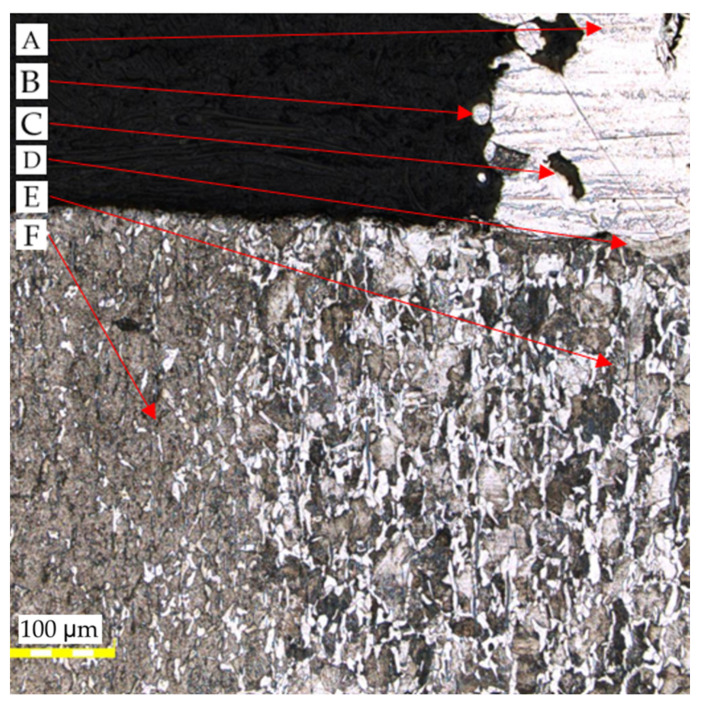
View of the parent and 3D-printed material with the visible connection between them: A—AM M300 steel; B—unmelted powder particle; C—porosity; D—connection zone; E—parent material; F—heat-treated zone of the parent material.

**Figure 13 materials-16-03772-f013:**
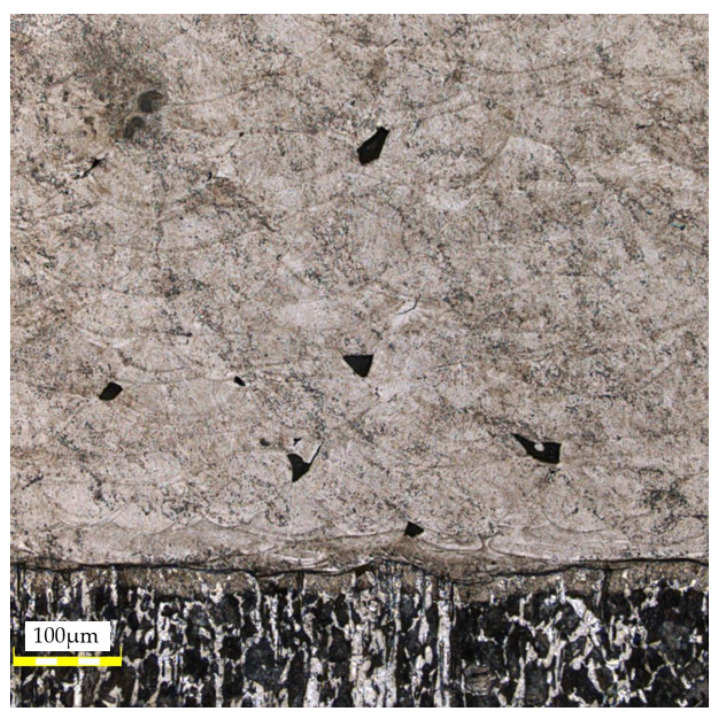
View of the connection between the 3D-printed material and the parent material.

**Figure 14 materials-16-03772-f014:**
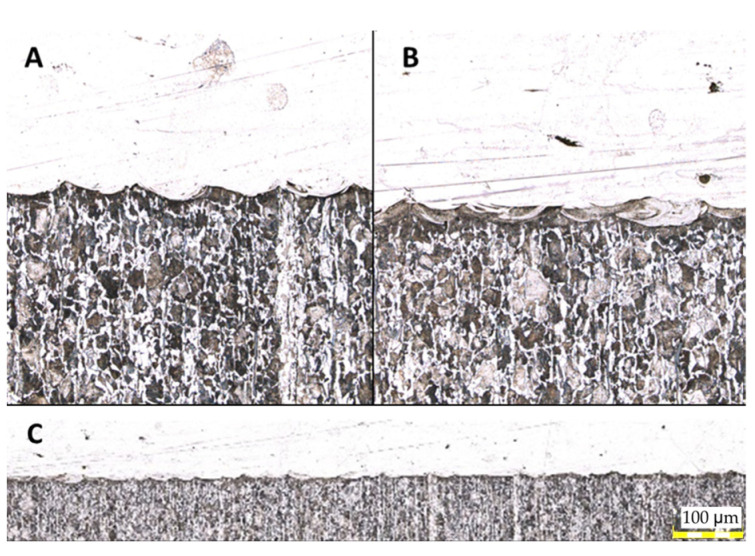
Structures illustrating the area of material joining: (**A**) without a visible material mix of two materials; (**B**) with a visible material mix of two materials; (**C**) the whole length of the connection.

**Figure 15 materials-16-03772-f015:**
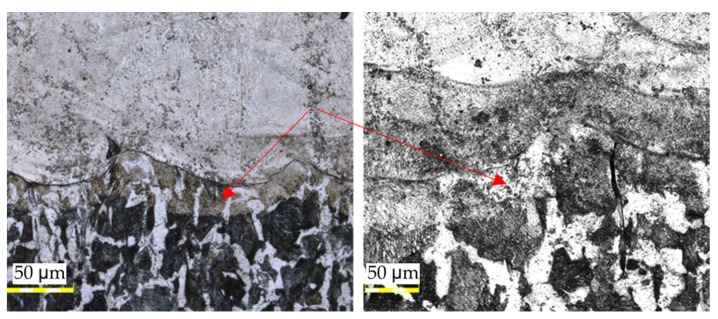
The structure of the transition zone (pointed by red arrows) is formed by regeneration between the native and regenerated material.

**Figure 16 materials-16-03772-f016:**
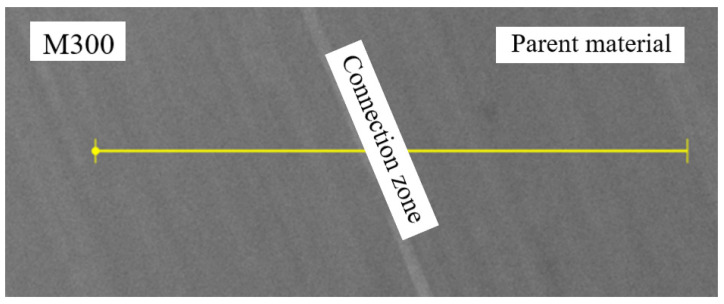
A schematic illustration of the measured segment to determine the chemical composition of the regenerated part.

**Figure 17 materials-16-03772-f017:**
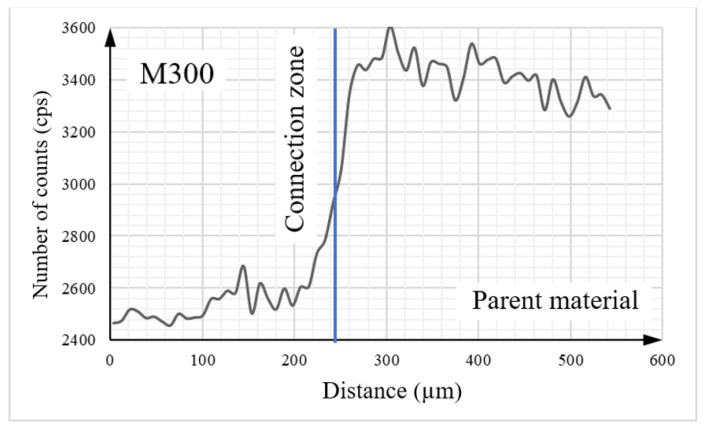
The graph illustrates the iron content in the tested sample.

**Figure 18 materials-16-03772-f018:**
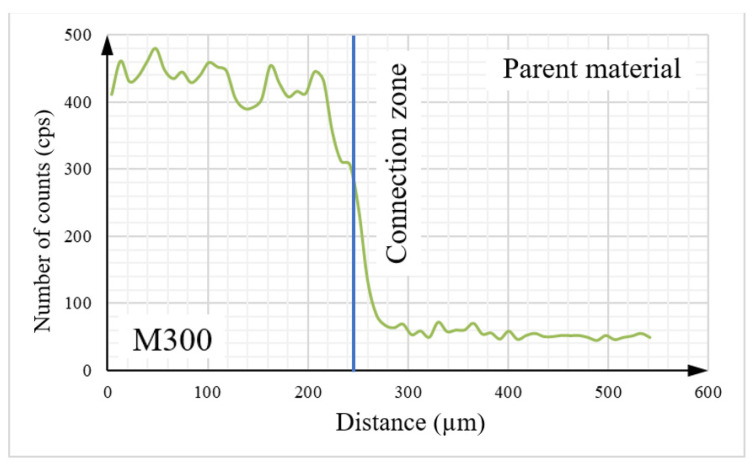
The graph illustrates the nickel content in the tested sample.

**Figure 19 materials-16-03772-f019:**
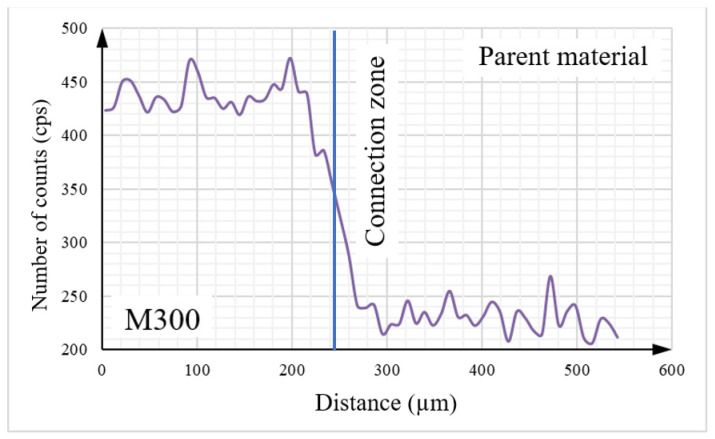
The graph illustrates the cobalt content in the tested sample.

**Figure 20 materials-16-03772-f020:**
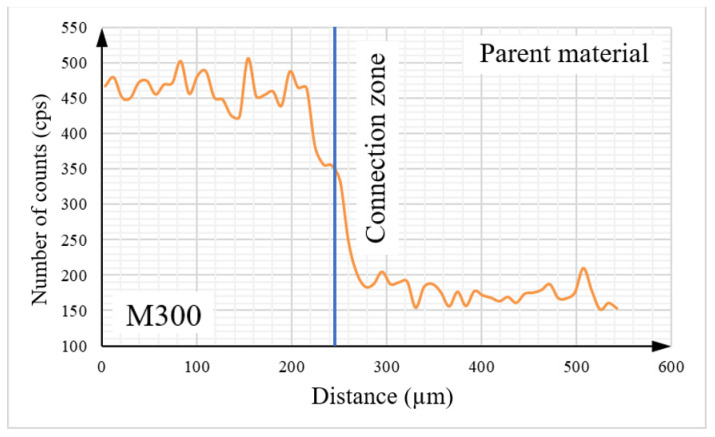
The graph illustrates the molybdenum content in the tested sample.

**Figure 21 materials-16-03772-f021:**
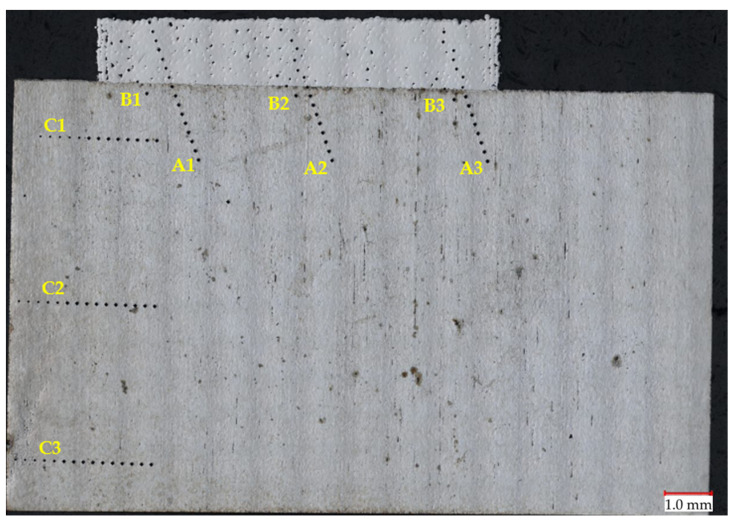
Diagram illustrating hardness measurement series: A1–A3—measurements of native material-M300 bonding; B1–B3—measurements of the transition zone; C1–C3—measurements of the heat-treated surface layer of the parent material.

**Figure 22 materials-16-03772-f022:**
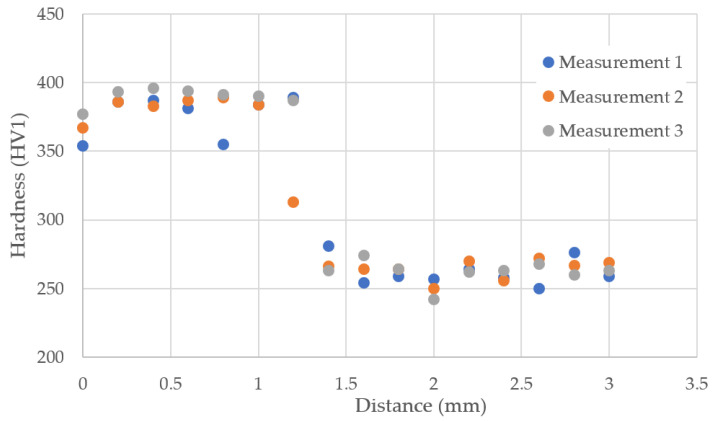
A hardness HV1 versus indentation distance plot was made on the joint between the native material and the regenerated material (M300).

**Figure 23 materials-16-03772-f023:**
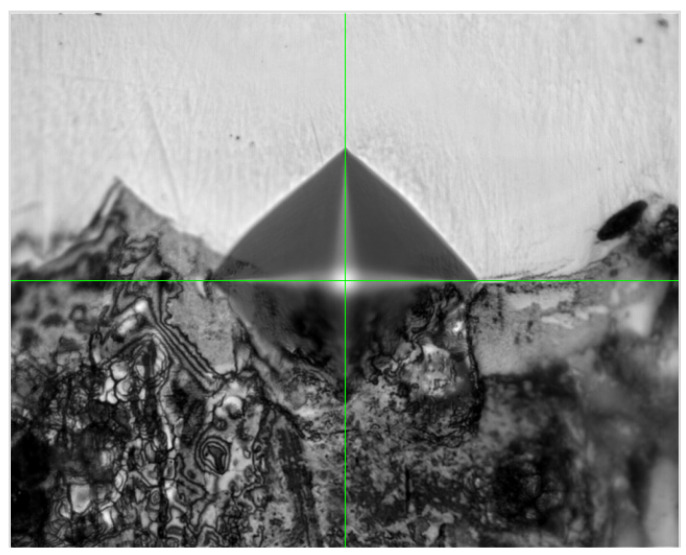
View of the indentation made centrally at the boundary of the materials.

**Figure 24 materials-16-03772-f024:**
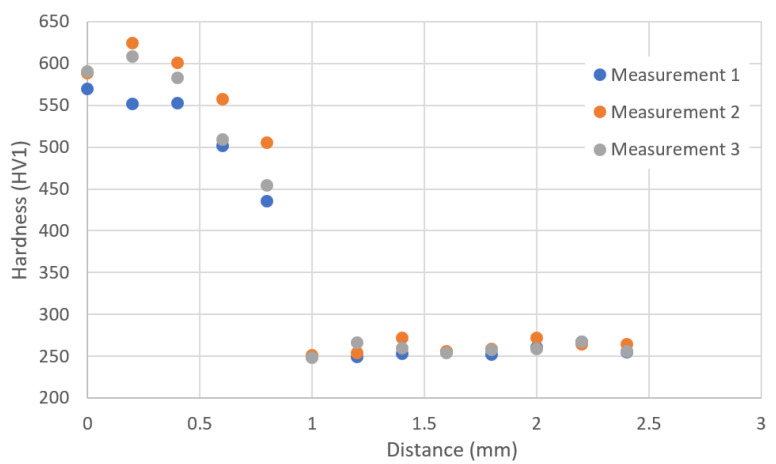
The graph shows the relationship between HV1 hardness and the distance of the indentation from the surface toward the core of the tested native material.

**Table 1 materials-16-03772-t001:** Chemical composition of the tested areas.

Area Type	Fe (%)	Mn (%)	Cu (%)	Si (%)	Ni (%)	Cr (%)	Co (%)	V (%)
A	98.05	1.11	0.27	0.25	0.15	n/a	2.07	n/a
B	64.52	<0.86	n/a	n/a	n/a	33.41	n/a	<0.20

**Table 2 materials-16-03772-t002:** Parameters used in the manufacturing process.

Parameter	Laser Power [W]	Exposure Velocity [mm/s]	Hatch Spacing [mm]	Layer Thickness [mm]	Energy Density [J/mm^3^]
Value	175.5	750	0.12	0.03	65

**Table 3 materials-16-03772-t003:** Chemical composition of P235GH steel and results from spectrometer investigation.

Source	Mn	Cu	Si	Ni
EN 10253-2	1.20%	0.30%	0.35%	0.30%
Investigation	1.06–1.16%	0.24–0.30%	0.21–0.29%	0.11–0.19%

## Data Availability

Not applicable.

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
