# Peer review of "Regeneration of the Damaged Parts with the Use of Metal Additive Manufacturing—Case Study"

_materials, 2023, doi:10.3390/ma16103772_

Round 1

Reviewer 1 Report

This study presents the results of tests conducted on a hydraulic distributor spool that was repaired using SLM. Comments are listed as follows:

1. In the abstract, the research background is too much, and the research conclusions need to be further condensed.

2.In the introduction, the author should explain the difference from other scholars' research and elaborate on the focus of this work. It is strongly suggested that the references need to make in-depth comments on the content of the cited papers; avoid generic comments. Mention/comment the relevance of the cited paper and especially the research gap associated to it. In addition, there are more relevant papers that should be covered in literature review:

https://doi.org/10.1016/j.surfcoat.2021.126964

https://doi.org/10.1016/j.jmapro.2019.12.009

3. The information in Figure 1 should be presented in a table

4.How many times have mechanical properties tests been repeated? This should be repeated at least three times

5.The accuracy of fracture images is not high enough, so scanning electron microscopy is required.

6.I don't think metallographic corrosion works well

English grammar should be improved.

Author Response

Dear Reviewer,

On behalf of all authors, I would like to thank you for taking the time to read our manuscript and put in your comments which allowed us to improve the quality of our work.  Regarding English improvements - the paper was proofread once again, and all changes are able to find because we used the "change tracking" option. Each correction based on your comments is yellow-highlighted in the manuscript. Below you can find our answers related to each of your comments:

  1. In the abstract, the research background is too much, and the research conclusions need to be further condensed.

    Ad.1. The abstract part has been rewritten based on your advice. 

  2. In the introduction, the author should explain the difference from other scholars' research and elaborate on the focus of this work. It is strongly suggested that the references need to make in-depth comments on the content of the cited papers; avoid generic comments. Mention/comment the relevance of the cited paper and especially the research gap associated to it. In addition, there are more relevant papers that should be covered in the literature review:

    Ad.2. We have improved the introduction part based on your comment, and added the mentioned citations. Please note that publication https://doi.org/10.1016/j.jmapro.2019.12.009
    has been cited in the original version of the manuscript (as [14]) - we also extended the description of this research. 
  3. The information in Figure 1 should be presented in a table

    Ad.3. The required table is added - see Table 1

  4. How many times have mechanical properties tests been repeated? This should be repeated at least three times

    Ad.4. The number of repetitions and proper statistical analysis of the obtained results are valid for standardized samples. In this case, we used a specific part. However, we made our tests on three sliders - the presented results are selected for the most objective and accurate test. 

  5. The accuracy of fracture images is not high enough, so scanning electron microscopy is required.

    Ad.5. We wanted to show the whole fracture area. Such a kind of merge of images is not possible to obtain via the SEM microscope that we have in our laboratory. In the case of magnification shown in Fig.10B, there are not any interesting phenomena to describe - there are only non-melted powder particles. 

  6. I don't think metallographic corrosion works well

    Ad.6. We are sorry but we cannot precisely understand your comment. If you mean the quality of the etched surfaces is not proper, please be informed we had to use different etching solutions, that were dedicated to each material, which is why it was not possible to keep the quality of both materials' microstructures at the same level. 

I hope our corrections would meet your expectations.

With kind regards,

Janusz Kluczyński

Reviewer 2 Report

The article “Regeneration of the damaged parts with the use of metal additive manufacturing – case study ” is dedicated to a study case of regeneration of the damaged parts with the use of metal additive manufacturingCongratulations on your work! I recommend revision with some improvements.

My recommendations:

1. In order to achieve a lower roughness of the slider surface further finishing processes were carried out on the regenerated part, as shown in Figure 3B. With whst? How?

2. Please do the axes from the graph more visible.

3. I would like to see DIC image of the reference slider compared with the regenerated one.

Author Response

Dear reviewer,

On behalf of all authors, I would like to thank you for taking the time to read our manuscript and put in your comments which allowed us to improve the quality of our work.  Each correction based on your comments is green-highlighted in the manuscript. Below you can find our answers related to each of your comments:

  1. In order to achieve a lower roughness of the slider surface further finishing processes were carried out on the regenerated part, as shown in Figure 3B. With whst? How?

    Ad.1. Thank you very much for finding this issue. We have rephrased this sentence. Now it has the following form: "In order to achieve a proper alignment for further tensile tests in uni-axial conditions, the produced part has been additionally machined, as shown in Figure 4B. "

  2. Please do the axes from the graph more visible.

    Ad.2. We analyzed the whole manuscript, and we found a Y-axis in Figure 8 with the visibility issues. Now it is improved. Thank you.

  3.  I would like to see DIC image of the reference slider compared with the regenerated one.

    Ad.3. Please see the attached file. We make that kind of comparison for you. At the same time, we did not put such a comparison in the manuscript because of the lack of any deformation during the corresponding stages of the tensile test. 
